# Attitudes toward Applying Facial Recognition Technology for Red-Light Running by E-Bikers: A Case Study in Fuzhou, China

Yanqun Yang [1], Danni Yin [1], Said M. Easa [2] and Jiang Liu [3,*]

1   College of Civil Engineering, Fuzhou University, Fuzhou 350008, China; yangyanqun@fzu.edu.cn (Y.Y.); yin-dn@foxmail.com (D.Y.)
2   Department of Civil Engineering, Ryerson University, Toronto, ON M5B 2K3, Canada; seasa@ryerson.ca
3   School of Architecture and Urban-Rural Planning, Fuzhou University, Fuzhou 350008, China
*   Correspondence: jiang.liu@fzu.edu.cn

**Abstract:** The application of facial recognition technology (FRT) can effectively reduce the red-light running behavior of e-bikers. However, the privacy issues involved in FRT have also attracted widespread attention from society. This research aims to explore the public and traffic police's attitudes toward FRT to optimize the use and implementation of FRT. A structured questionnaire survey of 270 people and 94 traffic police in Fuzhou, China, was used. In the research, we use several methods to analyze the investigation data, including Mann–Whitney U test, Kruskal–Wallis test, and multiple correspondence analysis. The survey results indicate that the application of FRT has a significant effect on reducing red-light running behavior. The public's educational level and driving license status are the most influential factors related to their attitudes to FRT ($p < 0.001$). Public members with these attributes show more supportive attitudes to FRT and more concerns about privacy invasion. There are significant differences between the public and traffic police in attitudes toward FRT ($p < 0.001$). Compared with the public, traffic police officers showed more supportive attitudes to FRT. This research contributes to promoting the application of FRT legitimately and alleviating people's concerns about the technology.

**Keywords:** facial recognition technology; e-biker; red-light running behavior; privacy invasion

## 1. Introduction

The e-bike is a vital means of transportation in many Chinese cities [1], given its convenience and fast characteristics. As of 2021, the number of e-bikes in China has reached nearly 300 million. The rapidly increasing number of e-bikes has resulted in increased accidents. In 2019, there were approximately 8639 deaths and 44,677 injuries caused by e-bike accidents, which is close to 70% of non-motorized vehicle casualties [2]. In China, e-bikes are categorized as non-motorized vehicles, and riders must drive on non-motorized lanes and comply with the same regulations as bicycles [3]. However, red-light running, illegal use of motor vehicle lanes, and over-speed cycling are the main reasons for accidents involving e-bikes [4]. These violations are often caused by low traffic safety awareness [5], among which running the red light is the leading cause of e-bike accidents [6,7]. Previous research points out that e-bikers run a red light more frequently than traditional bicycle riders [8], and e-bikes are faster than bicycles before collisions, with a higher risk ratio at intersections [9].

To reduce the red-light running behavior of e-bikers, many cities in China, such as Shenzhen, Shanghai, Jinan, and Fuzhou, have launched the Red-light Record System to regulate traffic violations. The system can capture and recognize the red-light running behavior of pedestrians and e-bikers and expose the screen's on-site violation images. The application of this system has achieved satisfying results. Since the Red-light Record System trial in Jiangbei, Chongqing, the violation rate of pedestrians and e-bikes has dropped from 40% to less than 3%. With facial recognition technology (FRT), traffic police need not face

the violators, and the difficulty of enforcement is reduced with the evidence provided from FRT. However, there is no specific law related to applying FRT in the traffic area. Thus, different cities have different standards for FRT. China has not yet established a unified standard for the application of FRT in transportation. The application of FRT has aroused public concerns about privacy invasion. Controversial opinions exist regard the extent to which violators' information is exposed and the suspicion around releasing personal privacy. Furthermore, whether it is a punishment beyond the law.

Thus, to understand the application effects of FRT, we investigate the attitudes of two significant stakeholders (the public and traffic police) on applying FRT in Fuzhou, China. The study aims to determine: (1) The public's opinion on the privacy violation of exposing personal information of red-light running behavior, (2) how personal characteristics of the public affect their attitudes toward FRT, and (3) the attitudes of traffic police toward FRT. Based on the above analysis, we propose several practical suggestions to improve the efficiency and rationality of FRT.

The methodology of the study is shown in Figure 1. The methodology consists of a literature review, experimental design, questionnaire design, data collection, and statistical analysis. In the analysis, all statistical calculations and plots were performed using SPSS 22.0.

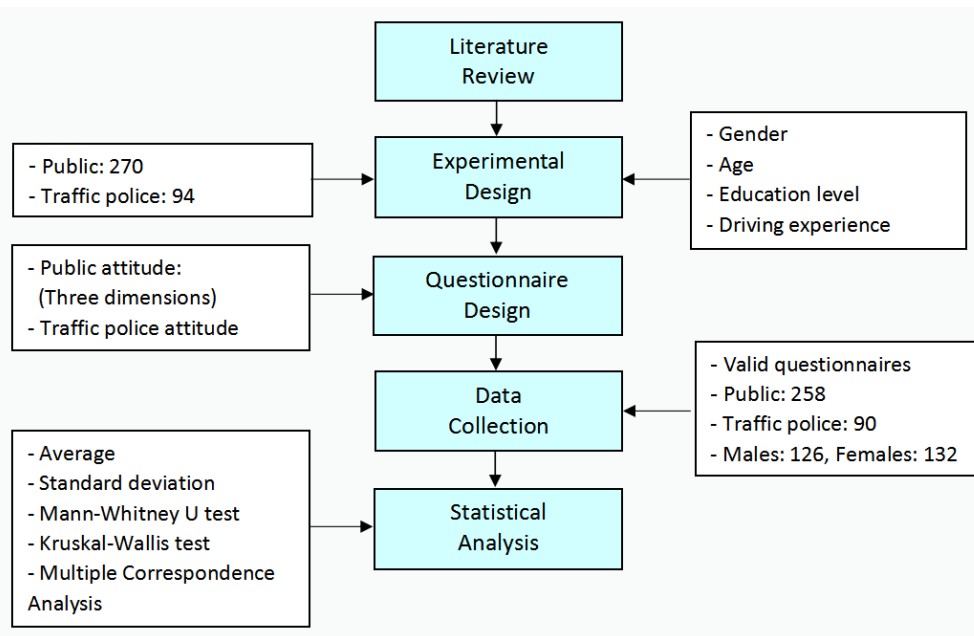

**Figure 1.** Study methodology.

## 2. Literature Review

### 2.1. Red-Light Running Behavior of E-Bikers

Many studies have been conducted to determine the factors that affect the red-light running behavior of e-bikers, mainly from external and internal perspectives. In terms of external factors, the higher acceleration rate and weight of an e-bike enables bikers to reach a higher speed than bicycle riding. Thus, e-bikers are more likely to run a red light [10,11]. Traffic conditions and situation factors have been verified to impact red-light running behavior [12]. They are also prone to accidents when the speed of an e-bike is underestimated by other road users [13]. As to internal factors, the attitudes of e-bikers are in close relationships with red-light running behavior. Red-light running intention and willingness could be predicted by the attitudes and past behaviors of e-bikers [14]. Self-discipline to follow traffic regulations, herd tendency, and past behaviors of e-bikers are crucial factors that affect the likelihood of accidents [15]. Of course, higher safety awareness and more concern about their traffic risk could reduce dangerous riding behaviors [16].

An acceptable waiting time for e-bikers at signalized intersections is shorter than that of bicycle riders, which may also be one reason for the higher probability of red-light running behavior [17]. Some scholars have found that gender and age may affect red-light running behavior. In terms of gender, males are more likely to run against a red light than females [18]. Although the effect of age on red-light running behavior is still not clear, young and middle-aged people are more likely to run a red light [19–21]. Whether holding a driving license or not could also affect the red-light-running violation rate [22]. To sum up, the complexity of these influential factors poses a significant challenge to red-light running behavior.

### 2.2. Preventive Measures

To prevent red-light running behavior, different intervention measures have been taken, such as educational programs, enforcement activities, and social marketing [23]. These interventions could use the positive influence of e-biker groups to promote law-obeying behavior [21]. Educating and training e-bikers is fundamental to reducing red-light running behavior [24]. E-bikers were recommended to participate in training programs to provide relevant skills [25]. Education and training programs for e-bikers with different characteristics reduce their unsafe behavior [26]. Besides, a comprehensive e-bike treatment needs enforcement [27]. Some scholars have recommended launching an e-bike license system with point-based penalties by factoring in China's unique regional and political characteristics [28]. Police enforcement of traffic regulations could effectively curb the red-light running behavior of e-bikers [24].

Technical equipment is widely used as an essential supplementary measure to monitor red-light running behavior. The equipment includes red-light cameras for motor vehicle drivers [29] and red-light running detectors performed by a system that consists of a camera and computer embedded in a motor vehicle [30]. Recognition systems using different technologies are used to monitor the red-light running behavior of cyclists and pedestrians. These technologies include video sequences, adaptive mapping techniques, and trained classifiers. Most of these technologies are related to image recognition. The video sequence is applied to detect red-light running behavior [31]. A real-time pedestrian recognition system that ensures high accuracy using a deep learning classifier and zebra-crossing recognition techniques is proposed using an adaptive mapping technique and a dual camera mechanism [32]. Finally, a recognition system for recognizing people at a pedestrian crossing is developed, which includes a trained classifier and two sets of images taken from an open database containing images of city streets from outdoor cameras [33]. These technologies can be used for image recognition of pedestrians and cyclists.

Among this technical equipment, FRT could be the most advanced one to monitor the red-light running behavior of pedestrians and non-motorized vehicles. These systems use FRT, including the red-light automatic early warning system and the red-light snapping system. The former is used for cyclists and pedestrians with automatic crossing reminders, red-light recording, exposure, and information inquiry [34,35]. In order to address the issue that the targeted face is subject to varying conditions, particularly of illumination, a novel pedestrian detection algorithm with multi-source face images is proposed [36]. With the red-light snapping system, the tracking success rate is increased to 85%, and the number of simultaneous tracking reaches 25 people [37].

However, due to the sensitivity of biometric data and the heterogeneity and openness of the network environment, the privacy leakage of biometric data is difficult to avoid [38]. Therefore, how to improve face recognition accuracy while ensuring high security of private data has provoked fierce public discussion.

### 2.3. Regulations and Privacy Concerns about FRT

Although advanced technologies could improve traffic safety, there are drawbacks at the same time. The main problem is the risk of privacy invasion since these technologies can collect, store, and share personal information [39]. For example, privacy and safety are

the main concerns expressed concerning traffic enforcement drones, and the citizens once opposed this technology in Los Angeles. They felt the department would use drones to track and observe them [40]. Privacy concerns are also reflected in in-vehicle data recorders. This concern tends to hinder the acceptance of innovations [41].

There are limited studies on the application feedback of FRT in recognition of red-light running behavior. However, numerous studies have conducted public surveys about FRT application, indicating their concerns about privacy invasion. In many cases, their facial information is collected involuntarily [42], which may lead to undesirable results of intrusions of privacy [43]. The privacy concerns are affected by privacy control, which means giving users the autonomy to control their private information [44]. The legitimacy of FRT contributes to allay, deaden, or possibly circumvent privacy concerns. In other words, FRT with less legitimacy could heighten people's concerns about privacy [45]. FRT also raises concerns about control over personal information, where it is used, and the potential for misrecognition [46]. These concerns about privacy invasion that FRT may cause have attracted worldwide attention.

The application of FRT for legal regulation has become the focus of legislative protection in various countries. Many states in the US have issued several bills about FRT. Government agencies in the US are cautious about using FRT and focus on prohibitive regulations. For example, the Body Camera Accountability Act states that the operation of FRT with a camera is an invasion of personal privacy [47]. Non-governmental organizations in the US are more open to using FRT, and they allow the restricted use of FRT to a certain extent. For example, Illinois proposed the Biometric Information Privacy Act (BIPA) to regulate the collection, storage, use, retention, and destruction of biometric information, including facial feature information, through individual empowerment and enhanced obligations [48]. The EU also restricts the application of FRT strictly. The General Data Protection Regulation (GDPR) incorporates different types and properties of personal information and protects personal information through civil, administrative, and criminal measures. In exceptional circumstances, the processing must meet the requirements of legal, legitimate, consent, and voluntary [49].

In China, the protection of facial features about FRT is distributed in laws and regulations. The Civil Code became effective on 1 January 2021, stipulating a natural person's personal information is protected, and the personal information mainly includes a name, birthday, and ID number. However, the Civil Code does not stipulate the contents and methods of protection expressly. China has announced more detailed regulations on the facial feature information involved in applying FRT in administrative regulations, rules, and other normative documents. Information Security Technology-Personal Information Security Specification revised in March 2020 explicitly regulates that personal biometric information is sensitive personal information. Sensitive personal information needs special protection. For example, before collecting personal biometric information, the subject should be informed of the purpose, method, and scope of personal information, storage time, and other rules, and the subject's consent should be obtained. Personal biometric information should be stored separately from personally identifiable information. In principle, original personal biometric information should not be stored. However, these regulations are only recommended and not mandatory [50]. The regulations about the application of FRT in China need to be further improved.

## 3. Questionnaire Design and Data Collection

### 3.1. Research Background

The research was conducted in Fuzhou, China. The application of FRT in Fuzhou, dates back to 2016 when the Fuzhou Traffic Police Department launched the first Red-light Record System at the intersection of Yangqiao Road and Daming Road. The system automatically can capture the images of the violators when they run a red light and recognize their personal information, and Figure 2 is the screen part of this system. Figure 3 is the red-light running behavior of e-bikers at the intersection. Then, the violators' mobile phones will

receive a message from the system, including the time and place of the violations. When the violators pay the fine, their images will disappear from the screen.

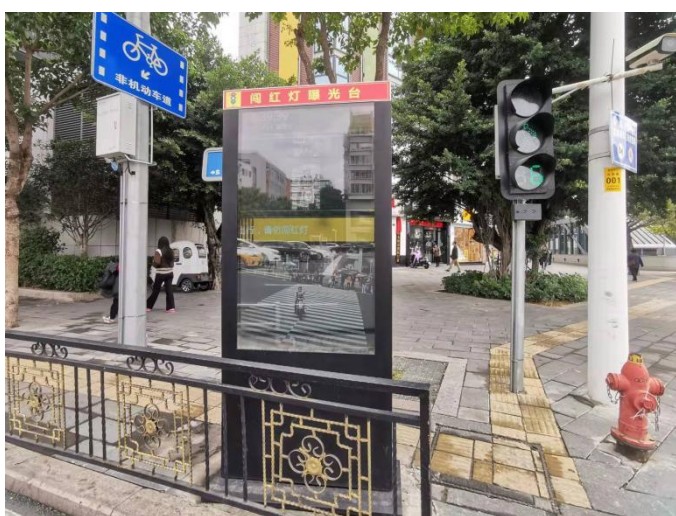

**Figure 2.** Part of the red-light monitoring system.

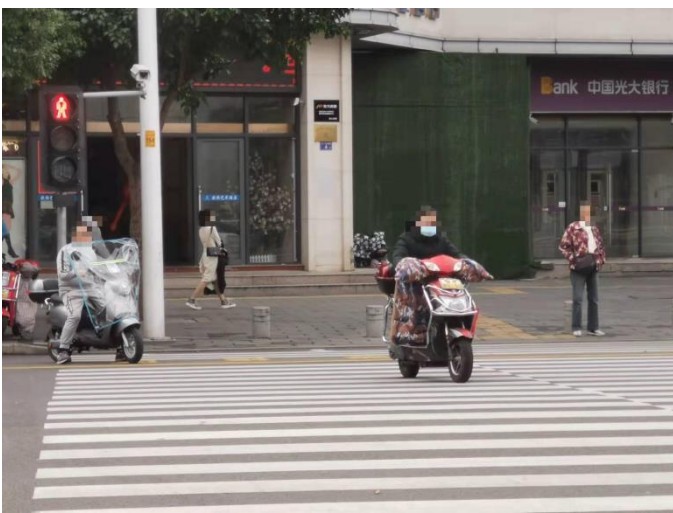

**Figure 3.** Red-light running behavior of e-bikers.

At the end of 2019, about 2.09 million e-bikes registered in the five districts of Fuzhou, China, resulting in increased regulatory difficulty. However, the application of FRT is facing contrary opinions. On the one hand, the effectiveness of FRT is recognized by part of the public who believe that FRT is more a deterrent than just a fine and by the traffic police for whom the technology substantially reduces the need for on-site supervision and provides reasonable evidence for punishment. On the other hand, some members of the public hesitate about accepting FRT as they are unsure whether their privacy is infringed and whether the collected information can be effectively protected.

### 3.2. Measures

3.2.1. Public Investigation

Referring to the Motor Vehicle Risky Driver Behavior Scale [51], and according to the characteristics of e-bikers and behavior, we designed a public investigation questionnaire. The questionnaire consists of two parts. The first part consists of basic personal information, including gender, age, education level, and driving license (Table 1). The second part is the public's attitudes toward FRT, which includes three variables: Attitudes toward red-light

running behavior, the application effect of FRT, and whether FRT violates privacy (Table 2). The first part of the questionnaire uses a single-choice form, and the second part uses a Likert five-level scale (from "strongly disagree = 1" to "strongly agree = 5").

**Table 1.** Items of basic personal information.

| Demographic Variables | Category |
|---|---|
| Gender | Male |
| | Female |
| Age | 18–36 |
| | 37–54 |
| | >54 |
| Education level | Junior high school and below |
| | Senior high school |
| | College and undergraduate |
| | Postgraduate and above |
| Driving license | Have |
| | Do not have |

**Table 2.** Survey items of public and traffic police attitudes toward FRT.

| Variables | Item |
|---|---|
| A: Attitudes toward red-light running behavior of e-bikers | A1: Red-light running behavior of e-bikers has a negative impact on traffic. |
| | A2: Even if I have good riding skills, running a red light may be dangerous. |
| | A3: Although running a red light can shorten the travel time, it is prone to accidents and is unworthy. |
| | A4: Red-light running behavior is irresponsible to lives. |
| | A5: More e-bikers are running red-light in China cities, and management needs to be strengthened. |
| | A6: I am familiar with the traffic regulations related to e-bikes, and I ride per the regulations. |
| B: Application effect of FRT | B1: FRT can significantly reduce the red-light running behavior of e-bikers. |
| | B2: The application of FRT helps to strengthen personal traffic safety awareness. |
| | B3: Need to take specific measures to punish the identified behavior. |
| | B4: The information of violators shown on the screen has a deterrent effect on the public. |
| | B5: Most people will support and actively obey the application of FRT in traffic management. |
| | B6: FRT is progress of technology and is worth promoting. |
| C: Whether FRT violates privacy | C1: It is not a violation of personal privacy to show red-light running behavior on the screen. |
| | C2: The following information published on the screen is appropriate: The offender's image running a red light (the face is covered), the middle part of the name is concealed, and the ID number is concealed in the middle digits. |
| | C3: Personal information identified by FRT will be strictly protected and will not be leaked. |
| | C4: The application of FRT also needs to be regulated by improving relevant laws. |
| | C5: When using FRT, the public's right to know needs to be guaranteed. |

**Table 2.** *Cont.*

| Variables | Item |
|---|---|
| P: Traffic police's attitudes toward FRT | P1: I have a good understanding of applying FRT to manage the red-light running behavior of e-bikers. |
| | P2: FRT can significantly reduce the red-light running behavior of e-bikers. |
| | P3: FRT can reduce the management difficulty of traffic police. |
| | P4: Need to take specific measures to punish the identified behavior. |
| | P5: Require to give safety education to the identified violators. |
| | P6: Most people will support and actively obey the application of FRT in traffic management. |
| | P7: The application of FRT helps to strengthen personal traffic safety awareness. |
| | P8: It is not a privacy violation to show red-light running behavior on the screen. |
| | P9: Personal information identified by FRT will be strictly protected and will not be leaked. |

### 3.2.2. Traffic Police Investigation

At the same time, we designed a questionnaire for traffic police from law enforcement officials' perspectives to understand their attitude towards FRT (Table 2). All questions use a Likert five-level scale (from "strongly disagree = 1" to "strongly agree = 5").

### 3.3. Participants

In July 2019, the questionnaires were distributed to the public and traffic police in Fuzhou, China. All ethical norms and standards were strictly followed during the survey. The survey randomly selected 270 people from the public. The requirements were: (1) They are between 18 and 70 years old and use e-bikes more than three times a week; (2) have lived in Fuzhou, China for more than 6 months, and (3) are able to understand and answer the questionnaire. Among the 270 public questionnaires, we excluded 12 partially unanswered questionnaires, and the remaining 258 questionnaires were valid. In addition, 94 traffic police officers in Fuzhou, China, were randomly selected. Four partially unanswered questionnaires were excluded, and the remaining 90 questionnaires were valid. Therefore, in the subsequent data analysis, only the valid questionnaires of public and traffic police are discussed.

### 3.4. Questionnaire Data Reliability

The test of the reliability and validity of the data set indicates that the Cronbach's $\alpha$ coefficient of the two questionnaires is greater than 0.7 [52], indicating good reliability of the questionnaires. Furthermore, the KMO and Bartlett spherical tests also meet the requirements of being greater than 0.6 with significance. Thus, the two questionnaire datasets used in this research are credible and compelling.

### 3.5. Demographic Data

Table 3 shows the demographic information of 258 interviewees of all valid questionnaires. The statistical results showed that the percentages of males to females surveyed are almost equal. Most of the people surveyed are in two age groups: 18–36 and 37–54. In terms of education level, most of them are with college and undergraduate degrees (43.4%) or high school degrees (33.7%), while other degrees account for a relatively low proportion. In addition, most of the interviewees have driving licenses (62.0%).

**Table 3.** Demographic information of the public.

| Demographic Variable | Category | Number | Percentage |
|---|---|---|---|
| Gender | Male | 126 | 48.8% |
| | Female | 132 | 51.2% |
| Age | 18–36 | 120 | 46.5% |
| | 37–54 | 114 | 44.2% |
| | >54 | 24 | 9.3% |
| Education level | Junior high school and below | 34 | 13.2% |
| | Senior high school | 87 | 33.7% |
| | College and undergraduate | 112 | 43.4% |
| | Postgraduate and above | 25 | 9.7% |
| Driving license | Have | 160 | 62.0% |
| | Do not have | 98 | 38.0% |

## 4. Analysis and Results

### 4.1. Statistical Analysis of Public Questionnaire Data

Table 4 shows the average and standard deviation of each item in the public questionnaire. From data statistics, the scores of the three variables are all between 3.2 and 3.3. Variable A has a score of 3.238, indicating the public generally regards running a red light to be dangerous behavior (A1 ~ A6), but two items of variable A (A5 and A6) have lower scores. The scores of variable A reflect that the public's awareness of observing traffic rules is relatively poor. The score of variable B is 3.278, indicating they are more supportive of the effect of using FRT (B1 ~ B6). Among items of variable B, only the scores of B2 are lower than the average scores, and the results reflect that FRT is less effective in improving the safety awareness of the public. Variable C has the highest score of 3.297, indicating they generally view that FRT does not violate their privacy (C1 ~ C5). However, C5 has the lowest score of 3.019, suggesting that the public is less concerned about the right to know the use of FRT. Above all, the public generally supports monitoring red-light running behavior by using FRT without worrying about privacy invasion too much.

**Table 4.** The average and standard deviation of each item in the public questionnaire, *n* = 258.

| Variable | Item | M | S.D. | Variable Average |
|---|---|---|---|---|
| A: Attitudes toward red-light running behavior of e-bikers | A1: Red-light running behavior of e-bikers has a negative impact on traffic. | 3.225 | 0.960 | 3.238 |
| | A2: Even if I have good riding skills, running a red light may be dangerous. | 3.256 | 0.782 | |
| | A3: Although running a red light can shorten the travel time, it is prone to accidents and is unworthy. | 3.302 | 0.865 | |
| | A4: Red-light running behavior is irresponsible to lives. | 3.318 | 0.784 | |
| | A5: More e-bikers are running red-light in China cities, and management needs to be strengthened. | 3.140 | 0.806 | |
| | A6: I am familiar with the traffic regulations related to e-bikes, and I ride per the regulations. | 3.186 | 0.849 | |

**Table 4.** *Cont.*

| Variable | Item | M | S.D. | Variable Average |
|---|---|---|---|---|
| B: Application effect of FRT | B1: FRT can significantly reduce the red-light running behavior of e-bikers. | 3.516 | 0.852 | 3.278 |
| | B2: The application of FRT helps to strengthen personal traffic safety awareness. | 3.012 | 0.766 | |
| | B3: Need to take specific measures to punish the identified behavior. | 3.287 | 0.848 | |
| | B4: The information of violators shown on the screen has a deterrent effect on the public. | 3.360 | 0.798 | |
| | B5: Most people will support and actively obey the application of FRT in traffic management. | 3.264 | 0.856 | |
| | B6: FRT is progress of technology and is worth promoting. | 3.229 | 0.836 | |
| C: Whether FRT violates privacy | C1: It is not a violation of personal privacy to show red-light running behavior on the screen. | 3.376 | 0.947 | 3.297 |
| | C2: The following information published on the screen is appropriate: the offender's image running a red light (the face is covered), the middle part of the name is concealed, and the ID number is concealed in the middle digits. | 3.384 | 0.853 | |
| | C3: Personal information identified by FRT will be strictly protected and will not be leaked. | 3.349 | 0.901 | |
| | C4: The application of FRT also needs to be regulated by improving relevant laws. | 3.357 | 0.907 | |
| | C5: When using FRT, the public's right to know needs to be guaranteed. | 3.019 | 0.996 | |

The research uses Mann–Whitney U and Kruskal–Wallis tests. The Mann–Whitney U test is used to explore: (a) Whether different genders and driving license statuses resulted in differences in the three variables regarding attitudes toward red-light running behavior of e-bikers, (b) determine the application effect of FRT, and (c) whether FRT violates privacy. There is no significant difference in terms of public's gender and three variables. However, there is a significant difference in terms of driving license status and the public's attitudes toward red-light running behavior (U = 998.000, $p < 0.001$) and the application effect of FRT (U = 2865.5, $p < 0.001$). However, there is no significant difference in the public's attitudes toward privacy invasion.

The Kruskal–Wallis test is used to explore whether the public's different ages and education levels resulted in differences in the three variables of public attitudes. There is no significant difference in terms of age and the three variables. There is a significant difference in terms of education level and the three variables, i.e., attitudes toward red-light running behavior of e-bikers ($\chi^2(3) = 114.730$, $p < 0.001$), application effect of FRT ($\chi^2(3) = 103.534$, $p < 0.001$), whether FRT violates privacy ($\chi^2(3) = 90.292$, $p < 0.001$).

Then, we use Multiple Correspondence Analysis (MCA) to study the correspondence between the public's characteristics and the three variables. In order to meet MCA's data requirements, the scope of variables (A, B, and C), values, and the classification values are shown in Table 5. Figure 4 is the joint plot of the category points. Table 6 and Figure 5 present the discrimination measures of the variables. The MCA transforms all variables of the original data through the optimal scale transformation to obtain two dimensions (Dimension 1 and Dimension 2).

**Table 5.** The scores of variables (A, B, and C) and the values after classification.

| Variable | Scope of Variable Value | Classification Values |
|:---:|:---:|:---:|
| A, B, C | (0, 1] | 1 |
| | (1, 2] | 2 |
| | (2, 3] | 3 |
| | (3, 4] | 4 |
| | (4, 5] | 5 |

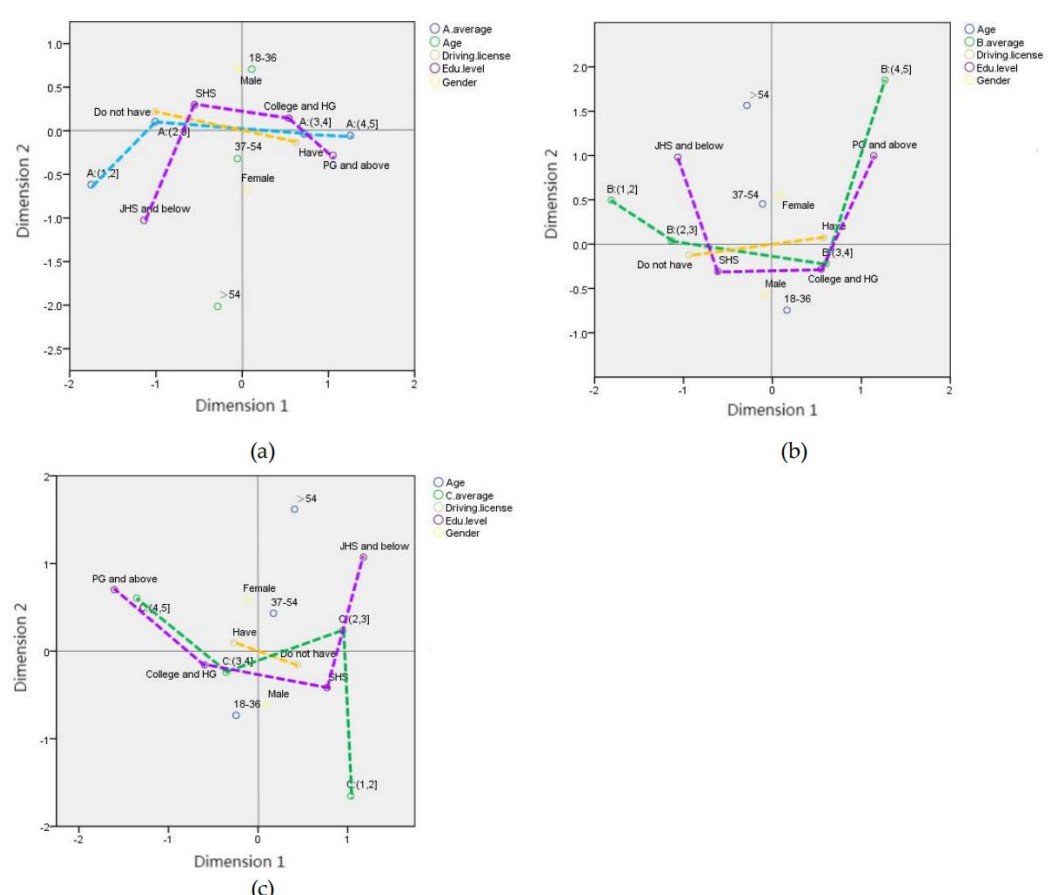

**Figure 4.** Joint plot of the category points. Correspondence between the variables: (**a**) The correspondence between the public's characteristics and variable A; (**b**) the correspondence between the public's characteristics and variable B; and (**c**) the correspondence between the public's characteristics and variable C. The education level in the figures is abbreviated (JSH and below stands for junior high school and below; SHS stands for senior high school; college and HG stands for college and undergraduate; PG and above stands for postgraduate and above).

From Figures 4 and 5, and Table 6, we could observe the correspondence between the public's characteristics and the three variables. In Figure 5 and Table 6, the public's education level (x = 0.509, y = 0.186) and whether holding a driving license (x = 0.630, y = 0.030) are related to the value of variable A, and the two characteristics also possess greater explanatory power to variable B (x = 0.533, y = 0.289; x = 0.541, y = 0.009). However, variable C only related to the public's education level (x = 0.787, y = 0.269).

**Table 6.** Discrimination measures of the variables.

| Variable | Dimension | | Variable | Dimension | | Variable | Dimension | |
|---|---|---|---|---|---|---|---|---|
| | 1 | 2 | | 1 | 2 | | 1 | 2 |
| Variable A | 0.889 | 0.017 | Variable B | 0.798 | 0.230 | Variable C | 0.679 | 0.196 |
| Gender | 0.003 | 0.483 | Gender | 0.007 | 0.310 | Gender | 0.011 | 0.362 |
| Age | 0.015 | 0.655 | Age | 0.025 | 0.577 | Age | 0.056 | 0.576 |
| Educ. Level | 0.509 | 0.186 | Educ. Level | 0.533 | 0.289 | Educ. Level | 0.787 | 0.269 |
| Driving License | 0.630 | 0.030 | Driving License | 0.541 | 0.009 | Driving License | 0.120 | 0.015 |

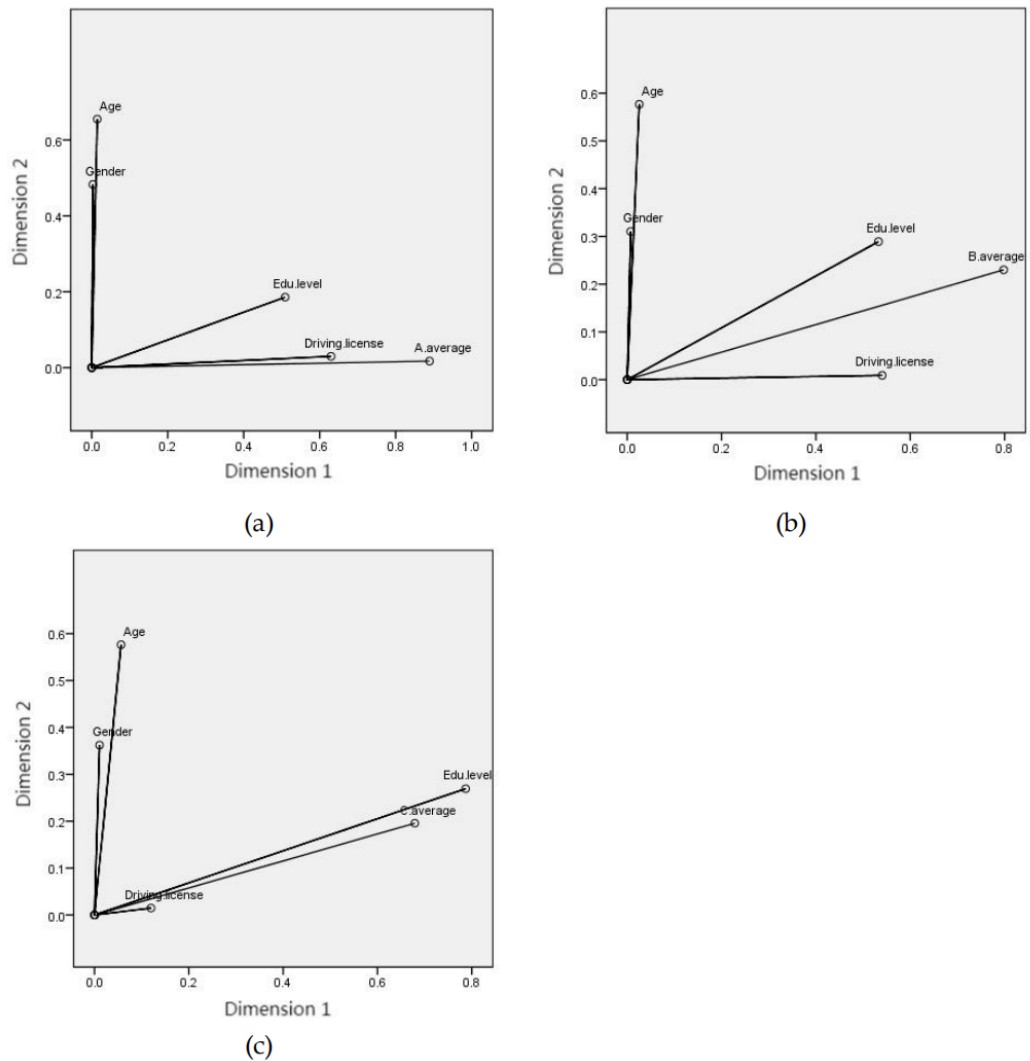

**Figure 5.** Discrimination measures of the variables: (**a**) Correspondence between the public's characteristics and variable A. (**b**) Correspondence between the public's characteristics and variable B. (**c**) Correspondence between the public's characteristics and variable C.

Figure 4 presents the points from the various categories. Different variables that are close to the same direction and the area of the graph may be related. In Figure 4a,b, the category points of the education level and whether holding a driving license are close to the specific scores of the variables A, B, and C. Specifically, the points of higher education level are closer to the higher scores of variables A and B. For example, "PG and above" is close to "A: (4, 5]" and "B: (4, 5]" and "College and HG" is close to "A: (3, 4]" and "B: (3, 4]".

In Figure 4c, "C: (1, 2]" and "JHS and below" have a long distance. The relationships between education level and variable C are similar to the situation in Figure 4a,b. In general, there are positive correlations between the driver's education level and the three variables. Besides, the points of whether holding a driving license are close to the points of variables A and B. Specifically, "Do not have" is close to "A: (2, 3]" and "B: (2, 3]" and "Have" is close to "A: (3, 4]" and "B: (3, 4]". However, whether holding a driving license does not have an obvious relationship with the points of variable C. The results indicate that people with a driving license get higher scores in variables A and B than those without a driving license. In short, whether holding a driving license positively affects variables A and B.

Table 7 and Figure 6 show the corresponding results between each of the three variables. In Figure 6b, the position of variable A (x = 0.752, y = 0.464) is close to that of variable B (x = 0.724, y = 0.419), and variable C (x = 0.305, y = 0.300) is farther than the two variables. In Figure 6a, the points position of the three variables with the same scores are also similar, except for the score (1, 2]. The results illustrate that the scores of the three variables have correspondence when the scores are higher. In other words, the scores of the three variables reach a higher level at the same time.

**Table 7.** Discrimination measures of the three variables.

| Variable | Dimension | |
|---|---|---|
| | **1** | **2** |
| Variable A | 0.752 | 0.464 |
| Variable B | 0.724 | 0.419 |
| Variable C | 0.305 | 0.300 |

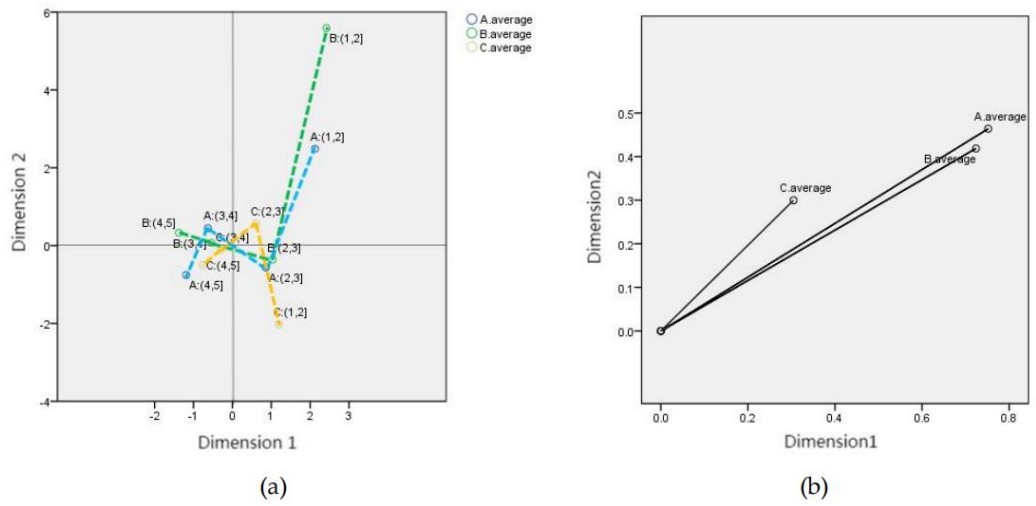

(a)  (b)

**Figure 6.** MCA results of the three variables: (**a**) Joint plot of the category points. (**b**) Discrimination measures of the three variables.

### 4.2. Statistical Analysis of Traffic Police Questionnaire Data

Table 8 shows the average and standard deviation of each item in the traffic police questionnaire. The results indicate that the average of most items is between 3.6 and 3.7, and the average of all items is 3.651. Among the traffic police questionnaire items, P7 has the lowest scores, which is like the public's results. Thus, from the perspectives of the traffic police, FRT can not entirely improve the safety awareness of e-bikers. Nevertheless, overall, the traffic police have high support for the use of FRT.

**Table 8.** Average and standard deviation of each item in the traffic police questionnaire, *n* = 90.

| Item | Item Average [1] | Standard Deviation |
|---|---|---|
| P1: I have a good understanding of applying FRT to manage the red-light running behavior of e-bikers. | 3.656 | 0.791 |
| P2: FRT can significantly reduce the red-light running behavior of e-bikers. | 3.678 | 0.747 |
| P3: FRT can reduce the management difficulty of traffic police. | 3.656 | 0.733 |
| P4: Need to take specific measures to punish the identified behavior. | 3.700 | 0.781 |
| P5: Require giving safety education to the identified violators. | 3.600 | 0.712 |
| P6: Most people will support and actively obey the application of FRT in traffic management. | 3.689 | 0.709 |
| P7: The application of FRT helps to strengthen personal traffic safety awareness. | 3.589 | 0.759 |
| P8: It is not a privacy violation to show red-light running behavior on the screen. | 3.656 | 0.653 |
| P9: Personal information identified by FRT will be strictly protected and will not be leaked. | 3.633 | 0.767 |

[1] The average of all items is 3.651.

### 4.3. Comparative Analysis of Questionnaire Datasets

Extracting the same items in the two questionnaires, the Mann–Whitney U test was used to explore the attitude differences between the public and traffic police toward FRT. The test results in Table 9 illustrate the two groups differ in attitudes toward FRT (U = 6958.500, $p < 0.001$).

**Table 9.** Differences in attitudes toward FRT between the public and traffic police.

| Variable | N | M | Mann-Whitney U | Wilcoxon W | Z | *p* |
|---|---|---|---|---|---|---|
| Public group | 258 | 3.300 | 6958.500 | 40369.500 | −5.691 | <0.001 |
| Traffic police group | 90 | 3.657 | | | | |

Table 10 shows the Mann–Whitney U test results of the same items by public and traffic police groups. There is a significant difference in the same items of the two groups, including B2/P7, B3/P4, B5/P6, C1/P8, and C3/P9, but there is no significant difference in B1/P2. The average of the same items also has a significant difference between the public and traffic police. Based on the results, it is concluded that there is a significant difference between public and traffic police attitudes toward FRT in general.

**Table 10.** Mann–Whitney U test results of the same items by public and traffic police groups.

| Item | Public | | Traffic Police | | Mann–Whitney U | Wilcoxon W | Z | *p* |
|---|---|---|---|---|---|---|---|---|
| | **M** | **S.D.** | **M** | **S.D.** | | | | |
| B1/P2 | 3.516 | 0.852 | 3.678 | 0.747 | 10592.000 | 44003.000 | −1.329 | 0.184 |
| B2/P7 | 3.012 | 0.766 | 3.589 | 0.763 | 7165.000 | 40576.000 | −5.935 | <0.001 |
| B3/P4 | 3.287 | 0.848 | 3.700 | 0.785 | 8605.500 | 42016.500 | −3.905 | <0.001 |
| B5/P6 | 3.264 | 0.855 | 3.689 | 0.713 | 8450.000 | 41861.000 | −4.139 | <0.001 |
| C1/P8 | 3.376 | 0.947 | 3.656 | 0.656 | 10069.000 | 43480.000 | −2.011 | <0.05 |
| C3/P9 | 3.349 | 0.901 | 3.633 | 0.771 | 9568.500 | 42979.500 | −2.648 | <0.05 |
| Average | 3.300 | 0.526 | 3.657 | 0.429 | 6958.500 | 40369.500 | −5.691 | <0.001 |

## 5. Discussion

### 5.1. Public Attitude toward FRT

In general, the public supports using FRT to manage the red-light running behavior of e-bikers. To understand which public characteristics are related to the attitudes toward FRT, we analyzed the correlation between the four individual characteristics and the three variables using the method of the Mann–Whitney U test, Kruskal–Wallis test, and MCA.

The results of the Kruskal–Wallis test and MCA indicate that members of the public with higher education levels are more resistant to the red-light running behavior of e-bikers ($\chi^2(3) = 114.730$, $p < 0.001$; Figure 4a). This finding is consistent with Wang et al. [53]. Under-educated e-bikers lack safety knowledge [53], and people with higher education backgrounds comprehend more traffic safety knowledge [39,45]. Members of the public with higher education levels are supportive towards the application effect of FRT ($\chi^2(3) = 103.534$, $p < 0.001$; Figure 4b), and they also show the trust of privacy protection ($\chi^2(3) = 90.292$, $p < 0.001$; Figure 4c). Because of more safety knowledge, people with higher education pay more attention to red-light running behavior and highly support FRT, perhaps due to their greater acceptance of new technologies. Moreover, their acceptance of FRT affects the trust of privacy protection.

Regarding whether or not holding a driving license affects the public's attitudes toward red-light running behavior and FRT ($U = 998.000$, $p < 0.001$; Figure 4a), people with driving licenses appeared to be more resistant to red-light running behavior. This is because e-bikers with driving licenses have lower perceived behavioral control and higher moral norm than those without driving licenses [23]. Moreover, e-bikers with driving licenses are also more supportive of the use of FRT ($U = 2865.5$, $p < 0.001$; Figure 4b). The strong correlation between the attitudes toward red-light running behavior and the application effect of FRT may indicate that people with driving licenses are more supportive of FRT.

### 5.2. Comparison of Public and Traffic Police Attitudes on FRT

The traffic police generally support the application of FRT (the average of all items is 3.651). Comparing the results of the same questions in public and the traffic police questionnaires shows that there are significant differences between the two groups in many items ($U = 6958.500$, $p < 0.001$), including "raise safety awareness, support for FRT applications, privacy issues of FRT, and information protection". The support from the traffic police to FRT is significantly higher than that of the public.

For the traffic police, how to reduce the red-light running behavior of e-bikers has been a difficulty [54], and the appearance of FRT has solved the problem well [55]. Thus, reducing management difficulty may be the main reason why the traffic police support FRT. For example, Shenzhen started to use FRT in 2017, which had reduced the number of red light-running behavior at intersections from about 150 cases per hour to about 8 cases per hour within half a year [56]. Besides, FRT can realize real-time monitoring, which is difficult

for traffic police [57]. The application of FRT can protect traffic police from personal injury caused by violators [58].

*5.3. Measures to Protect Public's Privacy*

The public generally believes that FRT does not violate their privacy (the average score of variable C is 3.297), indicating that FRT is trustworthy for the public. However, the attitudes toward privacy violations differ in the education level of e-bikers ($\chi^2(3) = 90.292$, $p < 0.001$). Overall, highly-educated e-bikers have more confidence in privacy protection involved in FRT. Thus, under the circumstance that information can be completely protected, the public's concerns about privacy violation can be alleviated.

The privacy about personal data (e.g., facial images) consists of the right to control the access to and use of these data [59]. Regulation of the use of FRT is vital for privacy protection. In China, FRT used at signalized intersections ensures traffic safety and protects public interest. However, laws and regulations to standardize FRT use in China are still not complete. The official privacy-preserving policy could mitigate some of the privacy concerns which seem to be most troubling for the public, such as blurring people's faces, allowing officers to access only violation footage, and so on [40]. Besides, the public should be well informed about the facial recognition systems and should have consented to use these systems for the specific and justified purposes in question [59].

Updated technologies are conducive to privacy protection. For instance, FRT based on temporal features could preserve privacy [60]. A face recognition protocol, named PEEP is used to protect privacy by utilizing differential privacy [61]. The principal components of adversarial segmented image blocks can protect people's privacy and prevent the distinct face-related features of images from being easily extracted [62].

## 6. Conclusions

This research developed two questionnaires for the public and traffic police and analyzed their attitudes toward applying FRT and its effects and privacy issues. The results indicate that:

(1)  The public's attitudes toward FRT are related to two individual characteristics: Education level ($\chi^2(3) = 114.730$, $p < 0.001$; $\chi^2(3) = 103.534$, $p < 0.001$; $\chi^2(3) = 90.292$, $p < 0.001$) and driving license status (U = 998.000, $p < 0.001$; U = 2865.5, $p < 0.001$). The MCA results (Figure 4) show that a person with a higher education level or a driving license supports FRT.

(2)  There are significant differences between the public and traffic police in attitudes toward FRT (U = 6958.500, $p < 0.001$). Traffic police support FRT application more than the public, as the technology is conducive to reducing red-light running behavior of e-bikers and enforcement effort.

(3)  Based on data analysis, we make some suggestions about the application of FRT. Improving the education level and safety knowledge of the public helps to enhance their support to FRT under the circumstances that privacy is protected completely. There are also several suggestions about the use of FRT. For example, laws and regulations on applying FRT could protect the public from privacy invasion, updating the technology of FRT to protect information better, and that the public should consent before using these systems for the specific and justified purposes in question.

(4)  This research has some limitations. The questionnaire designed for the public is not comprehensive enough, and more detailed questions about privacy violations could be included in the future. In addition, this research is only based on the investigation of e-bikers in Fuzhou, China and e-bikers from other cities and other groups (e.g., pedestrians and bicyclists) may have different attitudes toward FRT.

This study investigates the application of FRT from the perspectives of the public and traffic police. We analyzed the rationality of the use of FRT in combination with the public's attitudes toward personal privacy invasion. Our research has a certain contribution to society and science. Based on the research results, we recommend that government

departments should carry out the following tasks for the public, including people with low education and without driving licenses: (1) Conduct safety education and training regularly, and (2) promote the fact that the final purpose of FRT application is to improve public security's awareness and publish information without violating privacy.

**Author Contributions:** Conceptualization, Y.Y. and J.L.; methodology, Y.Y. and J.L.; validation, Y.Y., J.L., and D.Y.; formal analysis, D.Y.; investigation, D.Y.; writing—original draft preparation, D.Y.; writing—review and editing, Y.Y., J.L., and D.Y.; visualization, J.L. and S.M.E.; supervision, J.L. and S.M.E.; All authors have read and agreed to the published version of the manuscript.

**Funding:** This research received no external funding.

**Institutional Review Board Statement:** No applicable.

**Informed Consent Statement:** No applicable.

**Data Availability Statement:** The study did not report any data.

**Acknowledgments:** The authors are grateful to four anonymous reviewers for their thorough and most helpful comments. They also thank the Fujian Traffic Police Corps for funding this research and to Shihua Ni from the Fuzhou Traffic Police Detachment for his valuable support of this study.

**Conflicts of Interest:** The authors declare no conflict of interest.

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
