# Peer review of "Attitudes toward Applying Facial Recognition Technology for Red-Light Running by E-Bikers: A Case Study in Fuzhou, China"

_applsci, doi:10.3390/app12010211_

Round 1

Reviewer 1 Report

    Multiple correspondence analysis seems to me a good method of analysis for study data However, the method is not clearly described. Furthermore, the interpretation of the outputs of this method (Joint plots) seems to me to be approximate. It is said which variables are related, but not in which way. For example, does having a driving license positively or negatively affect variables A and B? Or, does those with higher education believe that FRTs violate privacy or not? This does not appear in the discussion of the results.

Author Response

Thank you so much for your comment.

Reviewer 2 Report

Attitudes toward Applying Facial Recognition Technology for Red-Light Running by E-Bikers: Perspectives of Public and Traffic Police in Fuzhou, China

A brief summary

The paper aims to explore the public and the traffic police's attitudes toward FRT to optimize the use and implementation of FRT. A structured questionnaire survey of 270 people and 94 traffic police in Fuzhou, China, was used.

Comments

Strengths of the paper:

  1. Professionally laid out paper – both academic and practical, comprehensive, clear and solid in terms of descriptive content and proposed considerations / applications.
  2. Extensive review references (65) on the considered topic and a clear review of published research works presented.
  3. Well thought out methodology by using different approaches and methods.
  4. The proposed method seems to be innovative and contains well-known hints of originality.
  5. The authors have been found promising results.

Weakness of the paper:

  • The paper title is too long.
  • It is not clear how the numerical statistical analyses influenced the results summarized in the conclusions.
  • Almost half of the article is devoted to statistical analyses, subsequently reading the conclusions does not have a numerical confirmation of these statistical analyses which were correctly performed by the authors to quantify the results of the questionnaires.
  • Both in the abstract and in the conclusions it would be necessary to report some numerical results of the statistical analyses.
  • In the abstract it is not mentioned that these statistical analyses were carried out.
  • … no more weaknesses!

The overall merit of presented research works and findings is high and definitely worth publishing after incorporation the above minor suggestions.

Author Response

Thank you so much for your comment.

Reviewer 3 Report

Congratulations on a job well done.

Author Response

Thank you so much for your comment.

This manuscript is a resubmission of an earlier submission. The following is a list of the peer review reports and author responses from that submission.

Round 1

Reviewer 1 Report

The statistical analysis methods used are very simple. The data deriving from the questionnaires could be treated with different methods that graphically represent the correlations between the variables considered, such as the Multiple Correspondences Analysis or Association Rules, etc. Representing the results by means of a graph would make the reader a more immediate understanding.
Why are A, B and C called Dimensions in table 2? In my opinion it would be better to call them Variables.
The conclusions need to be improved, they are currently only a reminder of the results. The contributions this study makes to scientific knowledge is unclear. 

Reviewer 2 Report

The paper "Efficiency or Privacy: Attitudes toward Applying Facial Recognition Technology for Red-Light Running by E-Bikers" should be deeply revised. The title is not compliant with the content of the manuscript because it only presents the results of two questionnaires. The English language should be revised: several errors are in the text. In my opinion it is not a scientific work: it seems a statistical analysis about the application of Facial Recognition Technology to reduce red-light running behavior.

Reviewer 3 Report

The paper titled “Efficiency or Privacy: Attitudes toward Applying Facial Recognition Technology for Red-Light Running by E-Bikers" is based on a questionnaire survey of 270 people and 94 traffic police in Fuzhou city and aims to explore the public and the traffic police's attitudes toward facial recognition technology (FRT).

So, the objective of the paper is relevant for the journal.

The manuscript topics fit well to the journal scope.

The research was conducted with acceptable methodological and scientific rigor even if I see two serious shortcomings:

1) The objectives of the study were not clearly explained at the beginning (in the introduction and also in the abstract)

2) Consequently, the conclusions are not clear and the reader does not find the motivation for such an accurate study.

Regarding the conclusion "Regulating the FRT application could protect the public from privacy invasion" seems necessary a deep analysis on the international rules about privacy. These rules are different from country to countries and this fact would do a loss of interest in this research. I advise the authors to add some information on this topic.

Reviewer 4 Report

  • The topic of the paper is very interesting and current.
  • Please format the paper according to the journal's instructions.
  • Although Fuzhou is a famous city in China, it would be good to have the name of the country next to the name of the city.
  • The introduction should state whether the recording of faces (facial recognition technology) is legally regulated. As I understand it, there is no legislation at the moment.
  • The paragraph in lines 152 to 159 is identical to the paragraph from lines 163 to 170.
  • I would commend a detailed and well-systematized review of the literature.
  • I hope this is the result of formatting, not negligence while writing the paper.
  • Excluding the partially unanswered questionnaires, 258 valid questionnaires were reclaimed.
  • Please clarify this part, so that there is no confusion about the total sample and incomplete questionnaires. „Excluding the partially unanswered questionnaires, 258 valid questionnaires were reclaimed.“
  • Most often, the probability is not stated in the form p = 0.000, but as p <0.001. Please replace this.
  • Match the eyelets, the capital letter „P“ cannot be used. Also, it is not correct to say P = 0.000, P <0.05, if P <0.05 refers to the set significance limit.
  • Write in the Methodology sections, which software you used to process the data. Also, write whether you followed all ethical norms and standards during the survey.
  • All in all, a very interesting and significant study. In conclusion, it would be good to add a section of recommendations, so that readers can more easily understand the scientific and social contribution of your paper.